# Plastic Film Mulching Improved Maize Yield, Water Use Efficiency, and N Use Efficiency under Dryland Farming System in Northeast China

**DOI:** 10.3390/plants11131710

**Published:** 2022-06-28

**Authors:** Md Elias Hossain, Zhe Zhang, Wenyi Dong, Shangwen Wang, Meixia Liu, Enke Liu, Xurong Mei

**Affiliations:** 1Institute of Environment and Sustainable Development in Agriculture, Chinese Academy of Agricultural Sciences, Beijing 100081, China; elias.abot@gmail.com (M.E.H.); dongwenyi@caas.cn (W.D.); wansunwen@163.com (S.W.); liumeixia@caas.cn (M.L.); liuenke@caas.cn (E.L.); 2Key Laboratory of Dryland Agriculture, Ministry of Agriculture of the People’s Republic of China (MOA), Beijing 100081, China; 3Department of Agricultural Botany, Faculty of Agriculture, Sher-e-Bangla Agricultural University, Dhaka 1207, Bangladesh; 4Liaoning Academy of Agricultural Sciences, Shenyang 110161, China; chick409@126.com; 5State Key Laboratory of Hulless Barley and Yak Germplasm Resources and Genetic Improvement, Lhasa 850002, China; 6Key Laboratory of Agricultural Environment, Ministry of Agriculture and Rural Affairs of the People’s Republic of China (MARA), Beijing 100081, China

**Keywords:** plastic film mulching, maize yield, water use efficiency, nitrogen use efficiency, soil nitrogen pool, dryland farming

## Abstract

This 2-year field study analyzed plastic film mulching (PFM) effects on nitrogen use efficiency (NUE), and soil N pools under rainfed dryland conditions. Compared to no-mulching (NM, control), maize yields under PFM were increased by 36.3% (2515.7 kg ha^−1^) and 23.9% (1656.1 kg ha^−1^) in the 2020 and 2021 growing seasons, respectively. The PFM improved (*p* < 0.01) the water use efficiency (WUE) of maize by 39.6% and 33.8% in the 2020 and 2021 growing seasons, respectively. The 2-year average NUE of maize under the PFM was 40.1, which was 30.1% greater than the NM. The average soil total N, particulate organic N, and microbial biomass N contents under the PFM soil profile were increased by 22.3%, 51.9%, and 35%, respectively, over the two growing seasons. The residual ^15^N content (%TN) in soil total N pool was significantly higher (*p* < 0.05) under the PFM treatment. Our results suggest that PFM could increase maize productivity and sustainability of rainfed dryland faming systems by improving WUE, NUE, and soil N pools.

## 1. Introduction

Increasing yields of major cereals such as maize with minimum environmental cost is one of the prime challenges for future world food security. Maize (*Zea mays* L.) as a C4 plant, is more efficient in photosynthesis, adaptive and productive in dryland environments than other cereals such as rice and wheat. Therefore, maize is predominantly cultivated in arid and semi-arid agricultural regions of the world, particularly in the dryland farming areas of north China [1]. However, in rainfed agriculture systems, which cover approximately 80% of the world’s cultivated land and contribute about 60% of the total crop production, grain yield is significantly limited, mainly due to water scarcity and soil nitrogen deficiency (N) [2,3,4]. Moreover, maize growth in the temperate semi-arid dryland areas of China is suffering from climate change-induced seasonal drought and cold springs.

Over the last few decades, mulching technologies have significantly increased world crop production, especially in dryland farming systems [5,6]. Mulching improves crop productivity, primarily by increasing water storage and altering soil temperature [7]. This is because of mulches’ ability to increase soil water storage by reducing evaporation and changing the hydrothermal condition of topsoil layers [7,8]. In addition, mulching helps increase soil nutrient cycling, maintain SOC and microbial biomass, improve soil enzyme activity, enhance soil aggregate stability, reduce soil erosion, decrease weed infestation and soil respiration [9,10,11,12]. Although different materials (organic, inorganic, or synthetic biodegradable) can be used as mulch covers, plastic film mulching provides some core advantages over others. Organic mulches such as crop straws are cheap but they may not be available (due to use for ruminant feeding or biofuel). Moreover, the use of crop strew for spring mulching is restricted in temperate semi-arid farming areas of China, due to their chilling effects on topsoil, which could retard seed germination and early plant development. Biodegradable film mulches are considered environmentally friendly, but they are very expensive, less accessible, and may not be viable for rainfed farming systems due to high cost to return ratio [13]. On the other hand, plastic mulches such as plastic films are easily accessible, moderate cost, and efficient for increasing crop yields by improving crop water storage and soil temperature (e.g., topsoil temperature, SOM, etc.). Although recent reports raised the issue that residual plastic film from improper mulch management has detrimental effects on soil structure, water and nutrient transport, thereby reducing crop production and disrupting the agricultural environment, and causing a series of pollution hazards [13,14,15], appropriate plastic film mulching is reported to reduce soil respiration and greenhouse gas emissions from farmlands [10]. For such multifaceted benefits, plastic film mulching has been widely used in the cultivation of maize in dryland farming areas of north China over the last three decades, and in return, maize production improved significantly. However, how plastic film mulching affects soil nutrient pools, especially labile organic N pools (PON, particulate organic N; MBN, microbial biomass N; and DON, dissolved organic N) under rainfed semi-arid maize production ecosystem, is less reported. This is important, because those labile organic pools are an early indicator of treatment-induced changes in soil total N pools, and affect soil N storage and supply for plant acquisition (for details of soil N pools, follow Hossain et al. (2021)) [16].

Plant growth and development typically rely on soil nitrogen (N) uptake and assimilation [17]. Because, N is a major yield limiting factor in most dryland farming areas, crop productivity mostly depends on external N inputs. In the maize-based rainfed farming areas of north China, chemical N fertilization (often overdosed) has been practiced for the last few decades to attain higher crop yields. However, large doses and or continuous application of N fertilizer could be damaging to soil health, and can exert potential risks of environmental pollution from farm leakages such as leachate deposition (NO_3_^−^, NH_4_^+^, dissolved organic N, etc.) and gas emissions (NH_3_, N_2_O, etc.) [16,18,19,20]. Such uncertain impacts of chemical fertilizers could threaten current food production and future food security. To mitigate the negative impacts of chemical fertilizers, enhancement of crop N use efficiency (NUE) and reduction of farm nutrient leakages have been previously suggested [21]. Plastic film mulching for fertilized dryland farming systems is considered as a possible alternative for promoting agricultural production because plastic film mulching and nitrogen (N) fertilizer has synergistic effects on maize productivity in cold temperate semi-arid ecosystems [22]. However, their impacts on soil residual N utilization are still unclear. Monitoring the effects of plastic film mulching on the fate of fertilizer N residues utilizing advanced nutrient tracing techniques could provide some useful information for sustainable soil N management.

Numerous previous studies with plastic film mulching and N fertilization in semiarid areas of China examined the grain yield of various crops (e.g., maize and wheat) through impacts on crop water and N use efficiency, soil temperature, soil microbes, etc. [23,24,25]. However, information of plastic film mulching effects on soil N pools and especially those labile organic pools, as well as the fate of residual N in soil-plant ecosystem are scarce. In this two-year field investigation, we developed six micro-plots in an ongoing long-term experiment field with plastic film mulching and no-mulching (control) treatments under rainfed semi-arid maize-growing ecosystem in northeastern China. We applied ^15^N-urea fertilizer to monitor the fate of fertilizer N in the soil-plant components. We aimed to: (1) analyze the effects of plastic film mulching on grain yield, WUE, and NUE of maize; and (2) investigate the impacts of plastic film mulching on the distribution of soil total N and its labile organic and mineral N fractions in the whole soil profile.

## 2. Experimental Methods

### 2.1. Experimental Site

The micro-plot field experiment was established at the Fuxin Agricultural Environment and Farmland Conservation Experimental Station of the Ministry of Agriculture and Rural Affairs in Fuxin Country, Liaoning Province, China. Geographically, this northeastern Chinese agricultural area is located at 42°02′ N, 121°67′ E, and 186 m (610 ft) above sea level. The climate in this area is characterized as semi-arid continental monsoon with a mean annual temperature of 7.2 °C, the mean annual precipitation is 483 mm (mainly falls between April and September), and an average frost-free period of 135–165 days. Soils at this site are classified as Udalfs according to the USDA soil taxonomy system. The physical and chemical properties of the initial soil at the test site are shown in Table 1.

### 2.2. Experimental Design

The two-year (2020–2021) micro-plot experiment was first established on May 2020 with two treatments: (1) No-mulching (NM), and (2) Plastic film mulch (PFM) in a chemically fertilized single-cropping rain-fed maize growing ecosystem. The micro-plots were developed in an experimental field where PFM and NM treatments were maintained with 3 replications, for a total of 6 plots, and the area of each field plot was 56 m^2^ (length × width = 8 m × 7 m). A single micro-plot was established in each field plot with an area of 0.96 m^2^ (1.2 m × 0.8 m). The micro-plots were prepared by in situ tamping of aluminum plastic plates (length × width × height = 1.2 m × 0.8 m × 0.7 m, in situ tamping 0.6 m into soil, exposed soil surface 0.1 m) with minimum disturbance of the soil in the middle of the micro-plot area. For plastic film mulching treatment, each micro-plot area was covered with 0.008 mm thick black polyethylene film. To conserve soil moisture for maize seed germination, the spring mulching was filmed on 28 April 2020 and 30 April 2021, 7 days before each sowing. Before mulching, the field was prepared by a 0.2 m deep plough. The spring maize cultivar (*Zea mays* L. cv. ‘Zhengdan 958’) was sown at a depth of 0.05 m, 0.3 m × 0.4 m spacing, and a density of 60,000 plants ha^−1^, using a hole-sowing machine. A total of 6 plants were cultivated in every micro-plot area. Before sowing and during plot preparation, ^15^N-labelled urea (99 atom %) was applied to each micro-plot, and the fertilizer rate was 225 kg ha^−1^. The ^15^N fertilizer was broadcasted in the 0.05–0.1 m topsoil of the micro-plot area, and the upper layer was covered with soil. Each micro-plot received a basal fertilization of 150 kg P ha^−1^ year^−1^ from superphosphate and 75 kg K ha^−1^ year^−1^ from potassium sulfate. The soil of the NM treatment received all the same preparations as the PFM treatment except film mulching. The micro-plots were harvested on 23 September 2020 and 30 September 2021. All the plots received chemical pest control measures to control insect pests and diseases.

### 2.3. Plant and Soil Sampling

Each year, plants from each microplot were harvested at maturity to determine the grain yield. Soil samples were collected following the harvest of spring maize. Two 0.05 m diameter soil cores down to a 1 m soil profile from each micro-plot were sampled using a soil auger. Each soil core of the selected profiles was separated into successional sub-samples at a 10 cm depth interval. Two sub-samples from the same depth category of each micro-plot were mixed together to form a composite sample. After removing organic stubble, each composite sample was divided into two parts. One part was air-dried and kept for analysis of soil chemical properties, and the second part of fresh sample was passed through a 2 mm sieve and stored at 4 °C for biochemical analysis.

### 2.4. Analysis and Measurements of Soil Samples

The soil samples of the selected profile collected at final harvest were used to measure soil total N (STN), labile organic N fractions (PON, particulate organic N; MBN, microbial biomass N; and DON, dissolved organic N), mineral N (NO_3_^−^ and NH_4_^+^) contents. Soil total N (STN) and PON were determined using air-dried soil samples. For analysis of STN, soil samples were ground and passed through a 0.15 mm sieve. STN was determined following the Kjeldahl digestion–distillation procedure as recommended by Bremner and Mulvaney (1982) [26]. PON was measured using the procedure as described by Bronson et al. (2004) [27]. MBN was determined using the fumigation-extraction method of Vance et al. (1987) [28]. The DON content was determined using the procedures as described by Gigliotti et al. (2002) [29]. The mineral N (NO_3_^−^ and NH_4_^+^) content in soil was estimated using the KCl extraction method [30]. The details of soil analysis procedures are described in our previous work [16]. The ^15^N content of plant biomass and soil was determined by stable isotope ratio mass spectrometry.

### 2.5. Calculation Procedures

Maize grain yields (kg ha^−1^) were calculated using the following Equation (1).
(1)$$GY=\frac{GYM\cdot AH}{MA}$$
where *GY* is maize grain yield (kg ha^−1^), *GYM* is grain yield obtained from a microplot, *AH* is the one-hectare land area (10,000 m^2^), *MA* is the area of a single micro-plot (m^2^).

The water use efficiency (or water productivity) was measured as a ratio of grain yield to total seasonal evapotranspiration (*ET*, soil evaporation and plant transpiration). Evapotranspiration was estimated using the general water balance method. The spring maize was cultivated under rainfed conditions, so the soil water content and cumulative seasonal precipitation were used to measure crop evapotranspiration. In a normal crop year, plant structure accounts for less than 1% of the total seasonal evapotranspiration; therefore, it was neglected during the ET estimation. The *WUE* (kg ha^−1^ mm^−1^) was calculated using the following Equations (2)–(4).
(2)WUE=GY/ET 
(3)ET=P+ΔSWC
(4)$$\Delta W=W_{H}-W_{S}\;$$
where *GY* is the spring maize grain yield (kg ha^−1^), *ET* is the evapotranspiration (mm), *P* is the precipitation, *∆SWC* is the soil water content (mm) in the 0 to 1 m profile, *W_H_* is the soil water content (mm) in the profile at harvest, and *W_S_* is the soil water content (mm) in the profile before seeding. Soil water content (SWC) was monitored using EcH_2_O (Decagon Devices, Pullman, WA, USA) soil moisture sensors that were attached to a data logger from sowing to harvest.

Nitrogen use efficiency (*NUE*) was measured as a ratio of observed grain yield to fertilizer application rate using the following Equation (5).
(5)NUE=GY/N
where, *NUE* is nitrogen use efficiency of maize, *GY* is grain yield (kg ha^−1^), and *N* is nitrogen fertilization rate (kg ha^−1^).

The fertilizer driven ^15^N content in soil was calculated using the following Equations (6) and (7).
(6)$$NDFF\;\left(\%\right)=\frac{15N_{sample}\;15N_{natural}}{15N_{straw}\;15N_{natural}}\times 100\;\;\;$$
(7)$$NDFF=NDFF\left(\%\right)\times N_{sample}\;\;$$
where *NDFF (%)* is the percentage of the total N content derived from ^15^N-urea fertilizer, *15N_sample_* is the ^15^N abundance in soil sample, *15N_natural_* is the natural abundance of ^15^N in each component (0.3665%) that was not treated with ^15^N-urea, *15N_fertilizer_* is the ^15^N abundance in applied ^15^N-urea (99 atom%), *Nsample* is the total N content in sample, *NDFF* is the ^15^N content in each soil sample.

### 2.6. Statistical Analysis

All plant and soil parameters were presented on an oven-dried weight basis. Results were reported as the mean and standard error (SE). The data were analyzed and an independent-samples t-test was performed for analysis of variance (ANOVA) using SPSS Statistics 25.0 software package for Windows (SPSS Inc. IBM, Chicago, IL, USA). The test for normality indicated that the data have not significantly deviated from the normal distribution based on Kolmogorov–Smirnov (*p* 0.462) and Shapiro–Wilk (*p* 0.714). The main treatment effects on variable means were detected and compared using the least significant difference (LSD) at the 0.05 probability level. All tables and figures were prepared with the EXCEL 2013 software package (Microsoft Inc., Redmond, WA, USA).

## 3. Results

### 3.1. Grain Yield, Water Use Efficiency and N Use Efficiency of Maize

Plastic film mulching significantly increased (*p* < 0.05) grain yield, water use efficiency (WUE), and N use efficiency of spring maize, over the two growing seasons (Table 2). In 2020, the grain yield of maize under PFM was 9450.3 kg ha^−1^, 36.3% (2515.7 kg ha^−1^) greater than that of NM. In 2021, the grain yield of PFM was 8580.1 kg ha^−1^, 23.9% (1656.1 kg ha^−1^) higher than that of NM. PFM significantly increased (*p* < 0.01) water use efficiency (WUE) maize in both growing seasons (2020 and 2021). The WUE of maize under PFM in 2020 and 2021 was 13.8 kg ha^−1^ mm^−1^ and 14.5 kg ha^−1^ mm^−1^, respectively. Compared to NM, PFM increased WUE of maize by 39.6% (3.9 kg ha^−1^ mm^−1^) and 33.8% (3.7 kg ha^−1^ mm^−1^) in the 2020 and 2021 growing seasons, respectively. NUE of maize under PFM was 42.0 and 38.1 in the 2020 and 2021 growing seasons, respectively, which were 36.3% (11.2), and 23.9% (7.4) greater than those observed under the NM treatment.

### 3.2. Total N and ^15^N Content in Soil

In both growing seasons, plastic film mulching (PFM) treatment significantly improved the soil total N (STN) contents in topsoil depths compared to those under no-mulching (NM) treatment (Figure 1A,C). The average STN contents in the profile under PFM treatment were increased by 17.17% (0.105 g kg^−1^) and 18.69% (0.107 g kg^−1^), in cropping years 2020 and 2021, respectively. In 2020, the PFM treatment had the significantly higher (*p* < 0.05) STN contents in 0–10 and 10–20 cm topsoil depths, which were 15.16% (0.136 g kg^−1^) and 16.94% (0.137 g kg^−1^), greater than those under NM treatment, respectively. In 2021, significantly higher (*p* < 0.05) STN contents were observed in 0–30 cm topsoil depths under PFM treatment, which was an average 16.79% (0.14 g kg^−1^) greater when compared to NM treatment. We found 25.72% (0.11 g kg^−1^) increment in STN content at 60–70 cm profile depth under the PFM treatment which was also significant (*p* < 0.05) over the NM treatment. Although STN content in all selected profile depths were higher under PFM treatment, the STN variations between PFM and NM treatments were identical (*p* > 0.05) in 20–100 cm depths of the profile in 2020, and in 30–60 cm and 70–100 cm depths of the profile in 2021 (Figure 1C).

In 2020, soil ^15^N content (%TN) differences between PFM and NM treatments were non-significant (*p* > 0.05) in the selected profile depths except in 50–60 cm and 80–90 cm subsoil depths (Figure 1B). The NM treatment had 168.72% (1.57% TN) and 174.21% (1.25% TN) greater soil ^15^N contents in 50–60 cm and 80–90 cm subsoil depths, respectively, which were significant (*p* < 0.01) when compared to the soil ^15^N contents (%TN) in respective soil depths under the PFM treatment. In the following season (2021), soil residual ^15^N content (%TN) differences between PFM and NM treatments were significant (*p* < 0.05) across the profile depths except 50–60 and 70–90 cm subsoil depths (Figure 1D). Irrespective of profile depths, soil residual ^15^N contents (%TN) were higher with PFM treatment. The average soil residual ^15^N content (%TN) in the profile of PFM treatment was 0.35 which was 85.77% (0.19) greater when compared to the average soil residual ^15^N content (%TN) in the NM soil profile. In 0–10, 10–20, 20–30, 30–40, 40–50, 60–70, and 90–100 cm depths the soil residual ^15^N content (%TN) under PFM treatment were increased by 85.94% (0.43), 90.96% (0.34), 56.89% (0.18), 203.14% (0.26), 82.18% (0.13), 97.32% (0.08), and 74.74% (0.05), respectively. The soil residual ^15^N content (%TN) differences between PFM and NM treatments were highly significant (*p* < 0.01) at 0–10, 10–20, 30–40, and 60–70 cm depths of the profile (Figure 1B).

### 3.3. Labile Organic N Pools

In both growing seasons (2020 and 2021), plastic film mulching (PFM) significantly increased particulate organic N (PON) contents in 0–30 topsoil depths (Figure 2). In 2020, the average PON content in the soil profile under PFM treatment was 77.4 mg kg^−1^ which was 28.69% (17.3 mg kg^−1^) greater than the average PON content of NM treatment. Compared to the NM treatment soils, the PON contents in 0–10 cm, 10–20 cm, 20–30 cm, 30–40 cm, and 40–50 cm profile depths under PFM treatment were increased (*p* < 0.05) by 20.44% (33.1 mg kg^−1^), 51.51% (42.8 mg kg^−1^), 41.84% (28.6 mg kg^−1^), 44.92% (23.0 mg kg^−1^) and 35.43% (16 mg kg^−1^), respectively. In 2021, PFM treatment had an average 85.8 mg kg^−1^ PON in the profile, which was 46.37% (27.2 mg kg^−1^) greater when compared to that under NM treatment. The PON contents in 0–10 cm, 10–20 cm, 20–30 cm, 30–40 cm, and 40–50 cm profile depths of PFM treatment were 203.5 mg kg^−1^, 166.5 mg kg^−1^, 118.5 mg kg^−1^, 83.0 mg kg^−1^, and 68.6 mg kg^−1^, respectively, which were 32.2% (49.6 mg kg–1), 60.29% (62.6 mg kg^−1^), 41.96% (35.0 mg kg^−1^), 53.42% (28.9 mg kg^−1^) and 85.94% (31.7 mg kg^−1^) greater and significant (*p* < 0.05) than those depths of NM treatment (Figure 2). The PFM treatment also had 59.57% (18.4 mg kg^−1^), and 40.88% (11.6 mg kg^−1^) improvements (*p* < 0.05) in soil PON contents at 70–80 cm, and 90–100 cm profile depths, respectively.

At the final harvest of maize in 2020, the microbial biomass N (MBN) differences between plastic film mulching and no-mulching treatments were significant at 0–10 cm and 10–20 cm topsoil depths (Figure 3). The average MBN content and MBN contents in topsoil depths were higher under PFM treatment. The average MBN content of PFM treated soil profile was 55.8 mg kg^−1^, which was 23.59% (10.7 mg kg^−1^) greater than the average MBN content of NM soil profile. At 0–10 cm and 10–20 cm topsoil depths, MBN contents of PFM treatment were increased by 60.66% (66.3 mg kg^−1^), and 50.76% (36.9 mg kg^−1^), respectively. In 2021, PFM had significantly higher MBN content in 0–10 cm, 10–20 cm and 60–70 cm profile depths (Figure 3). The average MBN in PFM soil profile was 57.37 mg kg^−1^, 46.47% higher than the average MBN content of NM soil profile. The PFM treatment had 128.35% (129.2 mg kg^−1^), and 57.79% (40.3 mg kg^−1^) greater MBN contents in 0–10 cm, and 10–20 cm profile depths.

In 2020, the dissolved organic N (DON) contents under plastic film mulching (PFM) and no-mulching (NM) treatments varied significantly (*p* < 0.05) in 0–10 and 10–20 cm profile depths (Figure 4). In 0–10 cm and 10–20 cm profile depths, soils of NM treatment contained 217.7 mg kg^−1^, and 154.5 mg kg^−1^, which were 31.73% (52.5 mg kg^−1^) and 39.79% (44.0 mg kg^−1^) greater and significantly higher (*p* < 0.05) than those observed under PFM soils. The average NM treatment of NM treatment was also increased by 21.39% (20.8 mg kg^−1^), when compared to the average DON content in PFM soil profile. The treatments’ effects were equal (*p* > 0.05) on DON contents between 20 and 100 cm depths of the profile. However, at second harvest (crop year 2021), significantly higher DON contents were observed in 0–40 cm depths of the profile under PFM treatment (Figure 4). The average DON content in PFM soil profile was 90.4 mg kg^−1^, which was 32.57% (22.2 mg kg^−1^) greater than the NM soil profile’s average DON content. At 0–10, 10–20, 20–30 and 30–40 cm profile depths PFM treatment contained 214.8, 116.4, 91.2, and 85.8 mg kg^−1^, respectively, which were increased by 82.68% (97.2 mg kg^−1^), 41.9% (34.3 mg kg^−1^), 19.72% (15.0 mg kg^−1^), and 27.99% (18.8 mg kg^−1^) when compared to those under NM treatment.

### 3.4. Mineral N Pools

At the final maize harvest of 2020, the average NO_3_^−^ and NH_4_^+^ contents in the profile under NM treatment were 32.07% (11.7 mg kg^−1^) and 23.05% (0.7 mg kg^−1^) higher when compared to the average NO_3_^−^ and NH_4_^+^ contents in the profile under PFM treatment, respectively. The NO_3_^−^ concentrations in 0–10 cm and 10–20 cm topsoil depths under NM treatment were 100.1 mg kg^−1^ and 70.8 mg kg^−1^, which were 19.27% (16.2 mg kg^−1^) and 46.18% (22.4 mg kg^−1^) greater and significantly higher (*p* < 0.05) than those under PFM treatment (Figure 5). Compared to the PFM treatment, the NM treatment had 72.66% (17.6 mg kg^−1^), 23.28% (6.2 mg kg^−1^), 73.41% (17.3 mg kg^−1^), and 79.20% (18.7 mg kg^−1^) higher NO_3_^−^ contents in 50–60 cm, 70–80 cm, 80–90 cm, and 90–100 cm subsoil depths, respectively, which were also significant (Figure 5). The NH_4_^+^ contents in 0–10 and 10–20 cm topsoil depths under NM treatment were 8.9 mg kg^−1^ and 7.4 mg kg^−1^, respectively, which were 46.29% (2.8 mg kg^−1^) and 89.32% (3.5 mg kg^−1^) greater and significantly higher (*p* < 0.01) than those observed under PFM treatment (Figure 6). The NH_4_^+^ concentration variations between PFM and NM treatments were non-significant (*p* > 0.05) in 20–100 cm depths of the profile.

At the final harvest of 2021, NO_3_^−^ concentrations in 0–40 cm profile depths under PFM treatments were higher (*p* < 0.01) than those under NM treatment (Figure 5). The average NO_3_^−^ content of PFM soil profile was 18.8 mg kg^−1^, which was 78.81% (8.3 mg kg^−1^) higher than the average NO_3_^−^ content of NM soil profile. Compared to the NM soils, the PFM treatment had 281.06% (45.9 mg kg^−1^), 104.39% (14.4 mg kg^−1^), 75.78% (9.9 mg kg^−1^), and 131.93% (14.5 mg kg^−1^) greater NO_3_^−^ contents in 0–10, 10–20, 20–30, and 30–40 cm profile depths, respectively. The observed NH_4_^+^ variations between PFM and NM treatments were significant only in 0–10, 10–20, and 90–100 cm profile depths (Figure 6). The NH_4_^+^ contents in 0–10 cm and 10–20 cm topsoil depths of NM soils were 3.6 mg kg^−1^and 3.4 mg kg^−1^, respectively, which were 16.20% (0.5 mg kg^−1^) and 25.73% (0.7 mg kg^−1^) higher and significant (*p* > 0.05) when compared to those under PFM treatment. At 90–100 cm, PFM treatment had 23.71% (0.5 mg kg^−1^) greater (*p* < 0.05) NH_4_^+^ content than NM soil.

## 4. Discussion

### 4.1. Effect of PFM on Grain Yield and NUE of Maize

Plastic film mulching increased grain yield and NUE of maize compared to no-mulching treatment at the same N fertilizer application rate (225 kg ha^−1^). Previous studies from semi-arid dryland farming areas reported significant increases in grain yield and NUE when plastic film mulching with reasonable N fertilizer were applied [22,25,31,32]. In semi-arid dryland areas of northwestern China, where spring weather conditions are close to our experimental site, an average 43.1% increase in grain yield was reported from PFM treatment [23]. According to several studies, improved soil water, temperature conditions and nutrient availability, are the main mechanisms for higher grain yields under PFM [33,34]. Liu et al. (2014) and Gao et al. (2014) found increased root growth under plastic film mulching treatment [35,36]. Increased soil water availability can stimulate maize root growth and promote NUE by influencing plant nutrient uptake and aboveground biomass production [31]. In addition to soil water availability, PFM possibly had higher fertilizer N content in topsoil profile, which could affect NUE of maize by maintaining a relatively uniform supply of available N throughout the growing seasons, which could provide an important explanation for higher biomass accumulation, grain yield and NUE of maize under PFM. Furthermore, higher maize yield under PFM could also be affected by mulch-induced alteration of soil temperature because soil temperature has a direct influence on root growth, SOM decomposition, and nutrient availability [37]. In these temperate semi-arid regions, mean air temperature is sub-optimum during the early growth stages of spring maize (mid-April to mid-June). Therefore, plastic film mulch is typically applied to enhance topsoil temperature and ensure seed germination [38]. Wang et al. (2016) suggested that the soil surface temperature under plastic mulch can be increased by 2–7 °C in the early stages of maize growth [39]. Consequently, the plastic film mulching likely facilitated the early plant development by increasing topsoil temperature, which could have some effects on biomass accumulation and grain yield of maize. Previous evidence and our results of grain yield and NUE improvements of spring maize indicate that soils under plastic film mulching had better hydrothermal conditions and N supply than no-mulching treatment.

### 4.2. Effect of PFM on Water Use Efficiency (WUE) of Maize

Agricultural land and water resources are decreasing and the world population is increasing; therefore, improvement in crop WUE is a major concern for ensuring food security and sustainability [40]. In this study, plastic film mulching increased the WUE of spring maize over the two growing seasons (Table 2). These findings indicate that plastic film mulching is effective in improving the WUE of spring maize. Previous studies have also proven the significant effects of mulching in increasing the WUE of maize [35,40,41,42]. Plastic film mulching increases the WUE because it reduces soil water evaporation and increases plant transpiration [43,44]. By reducing water flow from deeper soil layers to the topsoil through vapor transfer and capillarity, mulching directly reduces soil water evaporation, which in turn helps to maintain a relatively stable level of soil water content in the topsoil [45,46]. Such stable soil water content likely enhanced plant productivity and WUE. Under rainfed conditions (as followed in this study), plastic film mulching improves rainwater harvest and facilitates rainwater infiltration through capillaries to low-lying areas, ensuring adequate soil moisture near the plants’ root-zone [47,48], which could be other reasons for increased plant productivity and WUE under PFM treatment. The no-mulching treatment had lower WUE in both seasons, possibly due to higher soil water evaporation and poor soil conditions. Higher soil evaporation could affect the water productivity of spring maize by reducing soil water supply in critical growth and developmental phases. At this experimental site, the average temperature during early spring (mid-April to mid-May) was less than 15 °C, which is suboptimal for maize seedling emergence and development. Because plastic film mulching can increase topsoil temperature through greenhouse effects [49,50]; mulched soil likely had relatively improved soil conditions for early plant development, which could also affect spring maize yield and thus WUE.

### 4.3. Effects of PFM on Soil N Pools and ^15^N Content in Soil Profile

Soil total N (STN) has significant influence on soil physicochemical and biological properties and therefore plays a vital role in the soil health, fertility, and productivity of agro-ecosystems [51]. On the other hand, soil labile organic fractions are more sensitive to soil management practices than stabilized nutrient pools such as STN, and are therefore considered as indicators of agro-ecosystems sustainability. In this study, plastic film mulching significantly improved soil total N (STN) contents in 0–20 cm topsoil depths when compared to no-mulching (NM) treatment. The particulate organic N (PON) and microbial biomass N (MBN) content in topsoil layers were also improved under plastic film mulching. Mineral N (NO_3_^−^ and NH_4_^+^) contents in the profile depths showed variations with maize developmental stages (data not shown); however, overall data indicated that soil profile under PFM contained higher mineral N, particularly NO_3_-N. The possible reasons for improved STN under PFM mulching are likely the long-term (>10 years) simultaneous plastic film mulching and N fertilization effects on soil N immobilization pathways, particularly in labile organic N pools. Previous studies in fertilized dryland farming systems of China found positive N balance under plastic film mulching which may be another reason for increased STN content in mulched soil profile. Moreover, higher production of maize root biomass under PFM treatment indicates that PFM treatment provided a suitable rhizosphere environment for belowground biomass developments by increasing availability of N and water, which could have some influence on N immobilization in labile organic pools. Previous studies suggested that increased root growth, exudation, and turnover, could contribute to soil microbial biomass growth by enhancing organic C availability for microbial metabolism in the profile [35,36]. Because root biomass accumulations were higher under plastic film mulching (data not shown), decompensation of root biomass likely had a greater contribution to the PON fraction under PFM treatment. As a result, higher particulate organic N (PON) and microbial biomass N (MBN) contents were observed in topsoil layers under PFM treatment than under no-mulching treatment, over the two growing seasons. We speculate that higher N availability together with improved root biomass production, soil moisture, and SOC under plastic film mulching supported higher soil N immobilization, thereby increasing N concentrations in microbial biomass and particulate organic matter fractions of soil. The higher PON contents in topsoil depths under PFM soil profile also suggest that long-term plastic film mulching might positively influence soil aggregate stability, which can increase N immobilization in topsoil stable macroaggregates, and thus can reduce microbial mineralization of N, during off-crop seasons. Altogether, these factors could increase STN, PON and MBN contents of topsoil profile under PFM treatment.

No-mulching (NM) treatment had increased DON contents in 0–20 cm topsoil depths at the final maize harvest in 2020, but had reduced DON contents in 0–40 cm profile depths at the final maize harvest in 2021. Dissolved organic N (DON) is a highly mobile, and labile N source for soil microbes [52]. Therefore, DON status could be influenced by soil active microbial biomass and also by leaching with percolating water from seasonal precipitations. The DON variations under PFM and NM treatments in 2020 and 2021 could have two possible explanations: (1) In 2020, PFM soil might have had a higher DON consumption by soil microbes than NM treatment soil because PFM induced increases in soil water and temperature can change the characteristics of soil microbes, which could stimulate microbial activity and increases consumption of soil dissolved organic matters (DOC and DON) [53,54]; and (2) Due to the high mobility nature of DON, substantial DON might have leached down to deep soil with percolating water from seasonal precipitations in 2021. Although, the exact reasons for higher DON in topsoil depths under PFM treatment in 2021 are unclear, the average DON contents of PFM soil profile (97.0 mg kg^−1^ in 2020 and 90.4 mg kg^−1^ in 2021) suggest that due to long-term plastic film mulching and equal basal N fertilization in the experiment field, dissolved organic N status under plastic film mulching possibly attained a relatively stable condition where net microbial DON mineralization and DON origin from root exudation maize and SOM decomposition were nearly equal.

As we applied ^15^N-urea once in 2020 during basal fertilization, we did not observe any improvements in ^15^N content in soil total N pool at the end of first growing season (2020) under PFM treatment. As expected, PFM had substantial improvements in residual ^15^N contents in total N pool across the selected soil profile. These results clearly indicate that PFM increases the residual fertilizer N retention capacity of soil. It is worth noticing that PFM also had substantially higher ^15^N contents in plant biomass and grain in 2021 (data not shown). These findings suggest that PFM treatment not only increases N use efficiency of maize but also improves fertilizer N immobilization in soil N pools and/or reduces soil N loss through pathways such as NO_3_^−^ leaching, NH_3_ volatilization, etc.

## 5. Conclusions

In this two-year micro-plot study, plastic film mulching in a fertilized rainfed dryland farming system significantly increased grain yield, WUE, and NUE of spring maize compared to no-mulching treatment. Plastic film mulching improved soil total N, (STN), particulate organic N (PON) and microbial biomass N (MBN) contents in topsoil layers. Although soil mineral N (NO_3_^−^ and NH_4_^+^) contents varied with seasonal variation, plastic film mulching treatment possibly contained higher concentrations of NO_3_^−^ in the profile during active growth phases of maize development. Plastic film mulching also increased fertilizer N residue retention in soil total N pool in the selected profile, thereby reducing the contribution of the fertilizer to soil N loss pathways. Our findings suggest that plastic film mulching could be a good method for spring maize cultivation in fertilized rain-fed dryland farming systems because it improved grain yield, water use efficiency, N use efficiency, and soil N pool.

## Figures and Tables

**Figure 1 plants-11-01710-f001:**
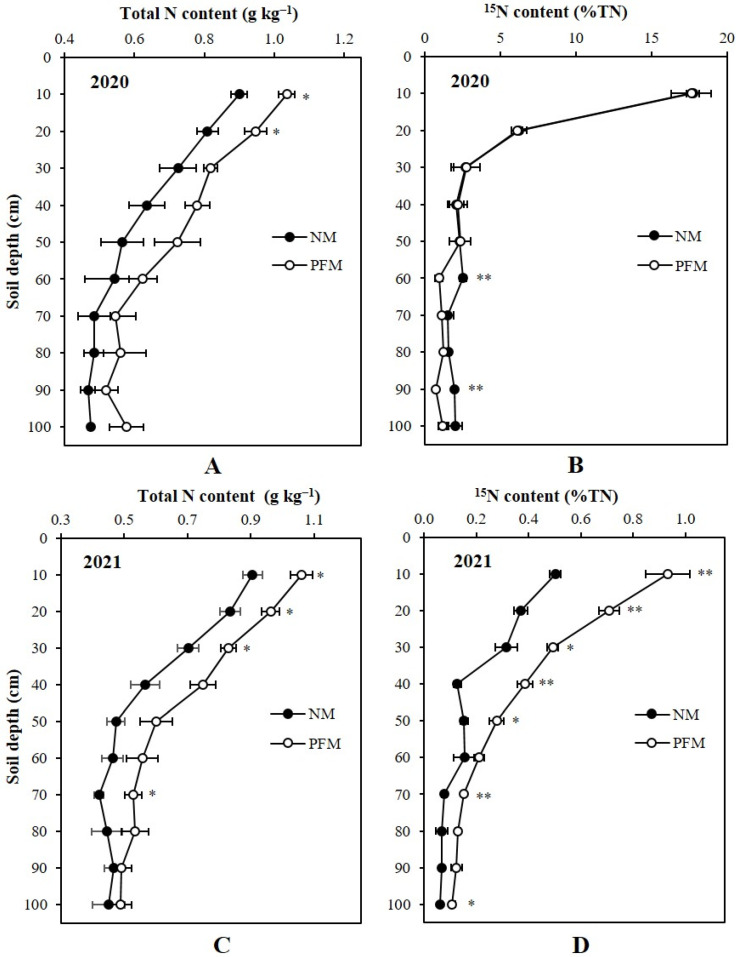
Effects of plastic film mulching and no-mulching treatments on the depth distribution of soil total N (**A**,**C**) and ^15^N content (**B**,**D**) in the soil profile. ** Significant differences between treatments at *p* < 0.01, * significant differences between treatments at *p* < 0.05. 2020, soil sampling at final harvest of maize in 2020; 2021, soil sampling at final harvest of maize in 2021. PFM, plastic film mulching; NM, no-mulching (control).

**Figure 2 plants-11-01710-f002:**
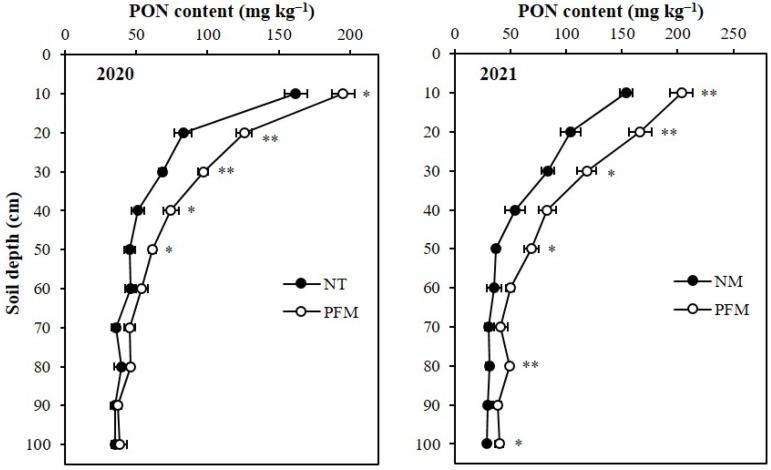
Effects of plastic film mulching and no-mulching treatments on the depth distribution of soil particulate organic N (PON) in the soil profile. ** Significant differences between treatments at *p* < 0.01, * significant differences between treatments at *p* < 0.05. 2020, soil sampling at final harvest in 2020; 2021, soil sampling at final harvest in 2021. PFM, plastic film mulching; NM, no-mulching (control).

**Figure 3 plants-11-01710-f003:**
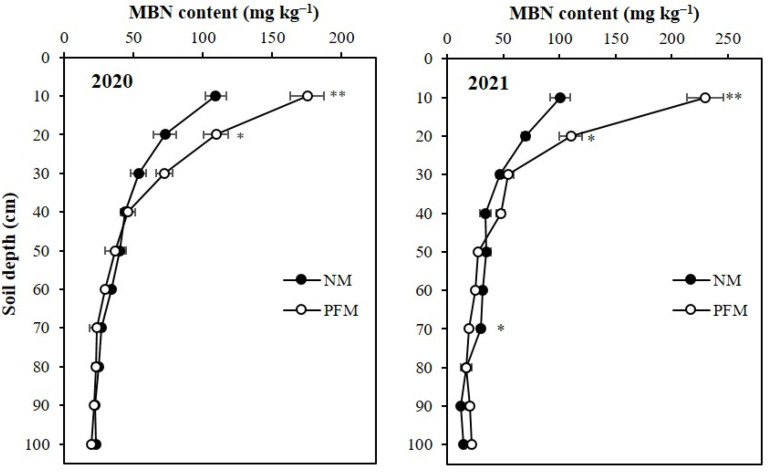
Effects of plastic film mulching and no-mulching treatments on the depth distribution of microbial biomass N (MBN) in the soil profile. ** Significant differences between treatments at *p* < 0.01, * significant differences between treatments at *p* < 0.05. 2020, soil sampling at final harvest in 2020; 2021, soil sampling at final harvest in 2021. PFM, plastic film mulching; NM, no-mulching (control).

**Figure 4 plants-11-01710-f004:**
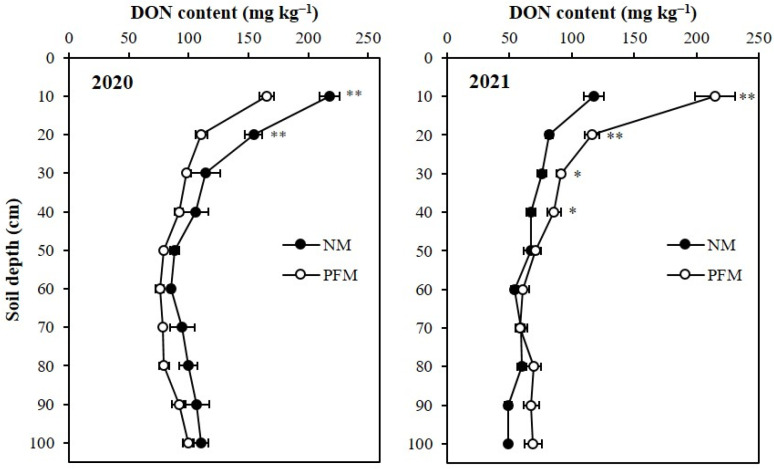
Effects of plastic film mulching and no-mulching treatments on the depth distribution of dissolved organic N (DON) in the soil profile. ** Significant differences between treatments at *p* < 0.01, * significant differences between treatments at *p* < 0.05. 2020, soil sampling at final harvest in 2020; 2021, soil sampling at final harvest in 2021. PFM, plastic film mulching; NM, no-mulching (control).

**Figure 5 plants-11-01710-f005:**
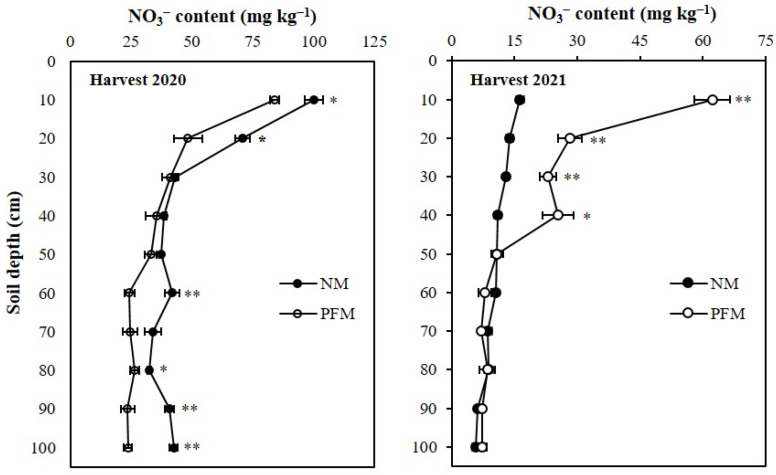
Effects of plastic film mulching and no-mulching treatments on the depth distribution of NO_3_^−^ in the soil profile at final harvest of maize. ** Significant differences between treatments at *p* < 0.01, * significant differences between treatments at *p* < 0.05. 2020, soil sampling at final harvest in 2020; 2021, soil sampling at final harvest in 2021 PFM, plastic film mulching; NM, no-mulching (control).

**Figure 6 plants-11-01710-f006:**
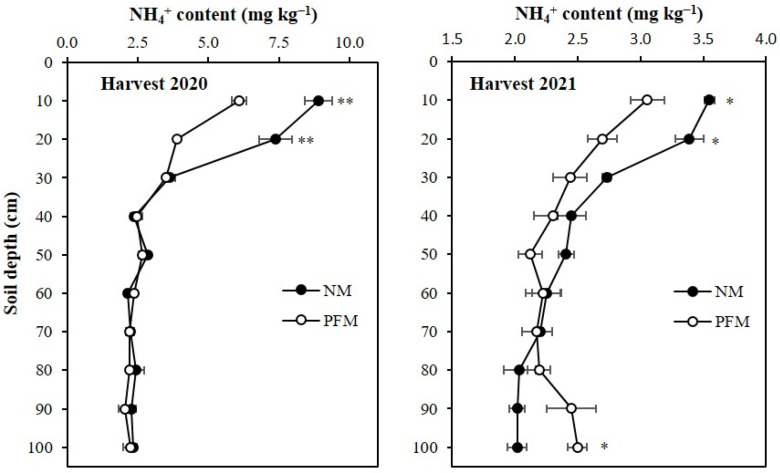
Effects of plastic film mulching and no-mulching treatments on the depth distribution of NH_4_^+^ in the soil profile at final harvest of maize. ** Significant differences between treatments at *p* < 0.01, * significant differences between treatments at *p* < 0.05. Harvest 2020, soil sampling at final harvest in 2020; Harvest 2021, soil sampling at final harvest in 2021 PFM, plastic film mulching; NM, no-mulching (control).

**Table 1 plants-11-01710-t001:** Physical and chemical properties of initial soil of the experimental site.

Soil Depth (cm)	pH	Organic Matter (g kg^−1^)	Total N (g kg^−1^)	Total P (g kg^−1^)	Total K (g kg^−1^)
0–20	6.95 ± 0.06	10.52 ± 0.39	0. 64 ± 0.02	0.66 ± 0.02	2.46 ± 0.08

**Table 2 plants-11-01710-t002:** Effects of plastic film mulching and no-mulching treatments on grain yield, water use efficiency, and nitrogen use efficiency of spring maize under rainfed semi-arid ecosystem of northeastern China.

Growing Season	Treatment	Grain Yield kg ha^−1^	Water Use Efficiency kg ha mm^−1^	N Use Efficiency kg kg^−1^
2020	NM	6934.6 ± 280.9 b	9.9 ± 0.4 b	30.8 ± 1.2 b
	PFM	9450.3 ± 493.2 a	13.8 ± 0.7 a	42.0 ± 2.2 a
	Significance	**	**	**
2021	NM	6924.0 ± 193.0 b	10.8 ± 0.3 b	30.8 ± 0.9 b
	PFM	8580.1 ± 490.9 a	14.5 ± 0.8 a	38.1 ± 2.1 a
	Significance	*	**	**

Note: Different letters in a column indicate a significant difference between treatments at *p* < 0.05. ** Significant difference between treatments at *p* < 0.01. * Significant difference between treatments at *p* < 0.05. PFM, plastic film mulching; NM, no-mulching (control).

## Data Availability

The data presented in this study are available on request from the corresponding author. The data are not publicly available because they belong to an ongoing project which has not finished yet.

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
