# Peer review of "Plastic Film Mulching Improved Maize Yield, Water Use Efficiency, and N Use Efficiency under Dryland Farming System in Northeast China"

_plants, 2022, doi:10.3390/plants11131710_

Round 1

Reviewer 1 Report

Please see comments in attachment

Author Response

Dear Reviewer, thank you for your great effort to improve our manuscript. Please see the attachment file for authors response to reviewers comments.   

Reviewer 2 Report

Minor comments and suggestions for authors are in the pdf file (attachment)

Author Response

Dear reviewer, thank you for your effort to improve our work. Please see the attachment for a point-by-point response to the reviewer’s comments.

Reviewer 3 Report

The manuscript is quite well written although it needs some wording checking. 

I miss some more discussion about the possible meaning of the increasing N content in depth under plastic mulching.

I include some comments within the manuscript highlighted with yellow marker.  

Author Response

Dear reviewer, thank you for your great effort to improve our work. please see Please see the attachment for a point-by-point response to the reviewer’s comments.
